# Structural basis of ClpXP recognition and unfolding of ssrA-tagged substrates

Xue Fei[1†], Tristan A Bell[1†], Sarah R Barkow[2], Tania A Baker[1], Robert T Sauer[1*]

[1]Departments of Biology, Massachusetts Institute of Technology, Cambridge, United States; [2]Chemistry, Massachusetts Institute of Technology, Cambridge, United States

**Abstract** When ribosomes fail to complete normal translation, all cells have mechanisms to ensure degradation of the resulting partial proteins to safeguard proteome integrity. In *Escherichia coli* and other eubacteria, the tmRNA system rescues stalled ribosomes and adds an ssrA tag or degron to the C-terminus of the incomplete protein, which directs degradation by the AAA+ ClpXP protease. Here, we present cryo-EM structures of ClpXP bound to the ssrA degron. C-terminal residues of the ssrA degron initially bind in the top of an otherwise closed ClpX axial channel and subsequently move deeper into an open channel. For short-degron protein substrates, we show that unfolding can occur directly from the initial closed-channel complex. For longer degron substrates, our studies illuminate how ClpXP transitions from specific recognition into a nonspecific unfolding and translocation machine. Many AAA+ proteases and protein-remodeling motors are likely to employ similar multistep recognition and engagement strategies.

*For correspondence:
bobsauer@mit.edu

†These authors contributed equally to this work

Competing interests: The authors declare that no competing interests exist.

## Introduction

ClpXP and related AAA+ proteases maintain cellular health by degrading incomplete, damaged, or unneeded proteins in a process that must be specific to avoid destruction of essential intracellular proteins (*Sauer and Baker, 2011*). ClpX (a AAA+ protein unfoldase/translocase) and/or ClpP (a self-compartmentalized peptidase) are also potential therapeutic targets for bacterial pathogens, such as *Mycobacterium tuberculosis*, as well as in human developmental defects, hematological disease, and cancer (*Bhandari et al., 2018*). In bacteria and eukaryotic organelles, AAA+ proteases typically recognize substrates via short N- or C-terminal peptide sequences. *Escherichia coli* ClpXP, for example, degrades proteins bearing a C-terminal degron called the ssrA tag that is added during tmRNA-rescue of stalled ribosomes (*Keiler et al., 1996*; *Keiler, 2015*). During rescue, tmRNA binds in the empty A-site of a stalled ribosome, adds a charged alanine to the nascent polypeptide in a tRNA-like reaction, replaces the original mRNA with a short open reading frame that directs translation of the remaining residues of the ssrA degron, and finally recruits translation termination factors via a stop codon. A different bacterial mechanism, which is similar to eukaryotic systems, adds alanine tails to the nascent polypeptide during the ribosome-rescue reaction (*Buskirk and Green, 2017*; *Lytvynenko et al., 2019*).

The sequence of the *E. coli* ssrA tag is AANDENYALAA-$\text{COO}^-$ (*Keiler et al., 1996*). The terminal Ala-Ala-$\text{COO}^-$ dipeptide of this degron is the most important element for ClpXP degradation (*Flynn et al., 2001*), and related degrons ending in Ala-Ala target other cellular proteins to ClpXP (*Flynn et al., 2003*; *Neher et al., 2006*; *Lytvynenko et al., 2019*). The ssrA tag initially binds in the axial channel of the hexameric AAA+ ClpX ring, where the pore-1, pore-2, and RKH loops contribute to recognition (*Siddiqui et al., 2004*; *Farrell et al., 2007*; *Martin et al., 2008b*; *Iosefson et al., 2015*). Subsequent mechanical reactions requiring ATP hydrolysis unfold adjacent regions of native protein structure and then translocate the denatured polypeptide through the channel and into the degradation chamber of the double-ring $\text{ClpP}_{14}$ peptidase for proteolysis (*Sauer and Baker, 2011*;

*Olivares et al., 2018*). Here, we establish the molecular basis of the recognition step in targeted ClpXP degradation of ssrA-tagged proteins. We also characterize subsequent unfolding/translocation steps that initiate processive degradation.

## Results

### Structure determination

For cryo-EM, we used an ssrA-tagged green fluorescent protein substrate (GFP-G$_3$YG$_9$SENYALAA; ssrA residues underlined), a single-chain *E. coli* ClpX$^{\Delta N}$ pseudohexamer (*Martin et al., 2005*), and *E. coli* ClpP. Approximately 15 s before vitrification, we combined the GFP substrate and ATP with a mixture of the ClpX$^{\Delta N}$ variant, ClpP$_{14}$, and ATPγS, which ClpX hydrolyzes more slowly than ATP (*Burton et al., 2003*). Classification and three-dimensional reconstruction of EM images containing one ClpX$^{\Delta N}$ pseudohexamer and one ClpP tetradecamer generated density maps of ClpX bound to one heptameric ring of ClpP$_{14}$ and 7–10 residues of the ssrA degron in two distinct conformations at resolutions of 3.1–3.2 Å (*Figure 1A and B*). Details of the cryo-EM workflow, refinement strategy, local resolution, and representative density for portions of ClpX or ClpP can be found in *Figure 1— figure supplements 1*, *2* and *3*, and in *Table 1*. As seen in previous high-resolution cryo-EM structures (*Fei et al., 2020*; *Ripstein et al., 2020*), subunits of the ClpX$^{\Delta N}$ hexamer formed a shallow spiral (labeled ABCDEF from the top to the bottom of the spiral in the clockwise direction), which docked asymmetrically with a flat ClpP$_7$ ring. The pore of the cis ClpP ring contacting ClpX was open, as expected (*Fei et al., 2020*; *Ripstein et al., 2020*), but the pore of the trans ClpP ring was closed in both new structures. The ssrA degron bound in the top of the ClpX channel in a structure we call the recognition complex and moved ~25 Å or six residues deeper into the channel in a structure we call the intermediate complex (*Figure 1B*). The presence of clear side-chain density for the ssrA degron is the most important difference between our current structures and those previously reported by our group (*Fei et al., 2020*) and by *Ripstein et al., 2020*. By contrast, in our previous structures, uncharacterized substrates apparently copurified with ClpXP, whereas a specific substrate was used for the studies of Ripstein et al. but no side-chain density was observed.

### Determinants of degron recognition

The ssrA degron bound high in the axial channel of the recognition complex, where it was contacted by the RKH, pore-1, and pore-2 loops of ClpX (*Figures 1B* and *2A–2E*; *Figure 2—figure supplement 1*). Importantly, access to the lower channel in this recognition complex was blocked by a previously unvisualized conformation of the pore-2 loop of ClpX subunit A (*Figures 1B*, *2B, C and D*). Blockage of the axial pore has not been observed in other structures of ClpXP or different AAA+ proteases and protein-remodeling machines. Thr$^{199}$ and Val$^{202}$ side chains from the blocking pore-2 loop in the recognition complex made van der Waals and hydrophobic interactions with the Cβ methyl of the C-terminal alanine of the ssrA degron, and the side chain –OH of the same Thr$^{199}$ side chain as well as the peptide –NH of Val$^{154}$ in the pore-1 loop of chain B made hydrogen bonds with the α-COO$^-$ of the ssrA degron (*Figure 2C and D*; *Figure 2—figure supplement 1*). Tyr$^{153}$ and Val$^{154}$ side-chain or main-chain groups from pore-1 loops (subunits A/B) or Arg$^{228}$ and His$^{230}$ side chains from RKH loops (subunits B/C/D/F) made additional contacts with the Ala-Ala-COO$^-$ or neighboring degron regions (*Figure 2C, D and E*; *Video 1*; *Figure 2—figure supplement 1*). As we discuss below, multiple experiments support the role of the ClpX-degron contacts observed in our recognition structure for specific binding of ClpX to the ssrA tag.

First, as expected based upon the recognition-complex structure (*Figure 2C and D*, *Figure 2— figure supplement 1A and B*), we found that mutation of the penultimate or ultimate C-terminal alanines of the ssrA tag in 29-residue peptide substrates increased $K_M$ for ClpXP degradation but had little effect on V$_{max}$ (*Figure 3A*). A prior study showed that mutation of the antepenultimate residue of the ssrA tag (leucine) or the amino acid two residues upstream (tyrosine) also increases $K_M$ modestly (*Flynn et al., 2001*), which can also be rationalized based upon recognition-complex contacts (*Figure 2C and D*, *Figure 2—figure supplement 1A and B*). Second, previously characterized Y153A, V154F, and R228A ClpX mutations increase $K_M$ for ClpP degradation of ssrA-tagged substrates 50-fold or more (*Siddiqui et al., 2004*; *Farrell et al., 2007*; *Martin et al., 2008b*; *Iosefson et al., 2015*). These mutational results are consistent with contacts observed between the

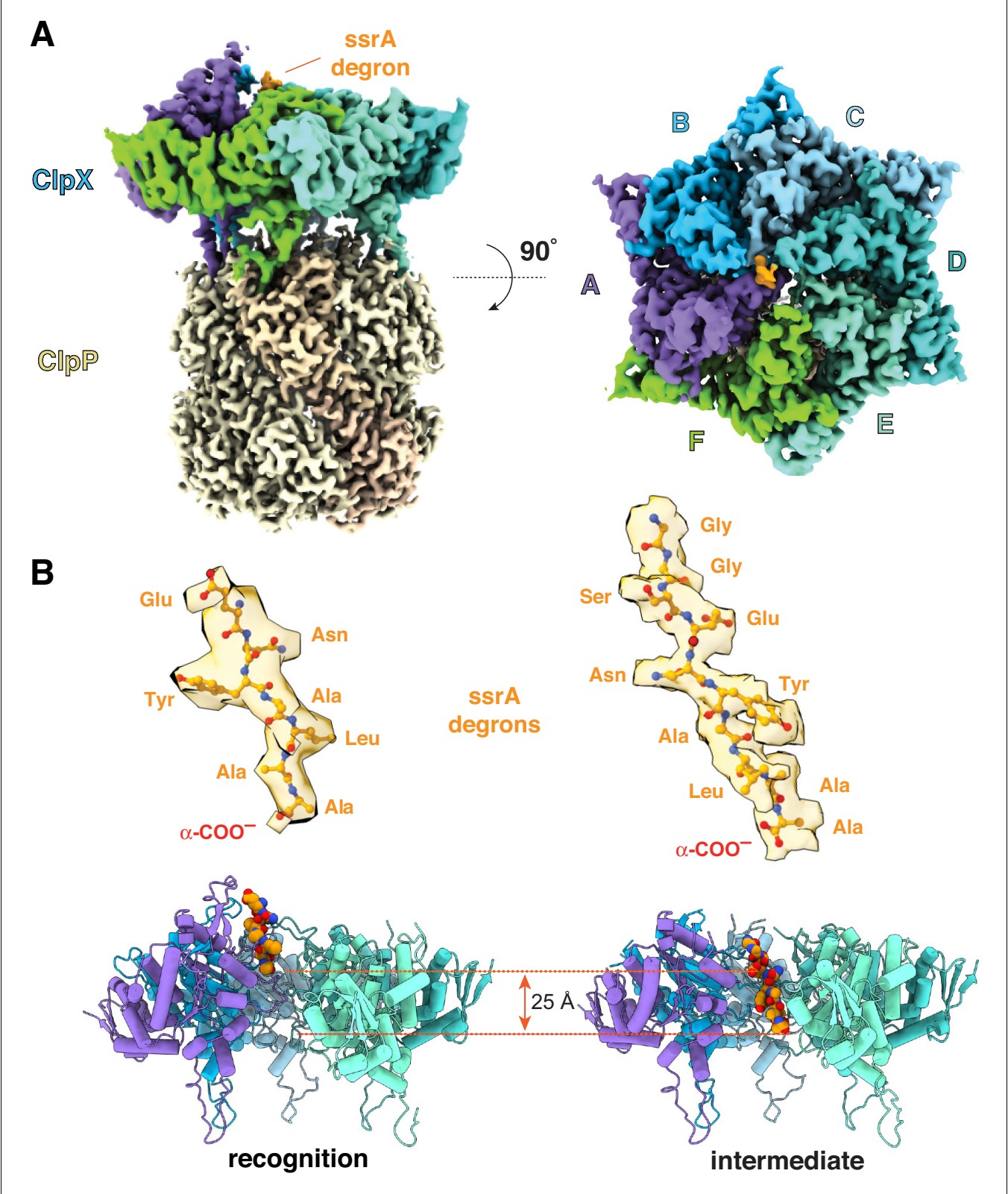

**Figure 1.** ClpXP complexes with ssrA degrons. (**A**) Side and top views of the composite cryo-EM density from the recognition complex. (**B**) The upper portion of the panel shows transparent density for the ssrA degron (stick representation) in the recognition and intermediate complexes; the lower portion shows the offset positions of the ssrA degron (space-filling representation) in the channel of ClpX (cartoon representation) in both complexes after removal of subunit F. In this and all subsequent figures, ClpX is colored blue, green, or purple; ClpP is yellow; and substrate is orange/gold.

*Figure 1 continued on next page*

*Figure 1 continued*

The online version of this article includes the following figure supplement(s) for figure 1:

**Figure supplement 1.** Cryo-EM data and strategy.
**Figure supplement 2.** Cryo-EM data validation.
**Figure supplement 3.** Representative density.

wild-type side chains and portions of the ssrA tag in the recognition complex (*Figure 2A and B*; *Figure 2—figure supplement 1A and B*). Based upon the recognition-complex structure, we also constructed new T199A, T199S, T199V, V202A, and H230A variants. In assays of GFP-ssrA degradation, the T199A, T199V, V202A, and H230A mutations caused large increases in $K_M$ (*Figure 3B*). By contrast, $K_M$ for degradation by the conservative T199S variant increased only ~4-fold (*Figure 3A*),

**Table 1.** Cryo-EM data collection, processing, model building, and validation statistics.

| Name | ClpP | ClpXP-ssrA Recognition complex | ClpXP-ssrA Intermediate complex |
|---|---|---|---|
| PDB ID | 6WR2 | 6WRF | 6WSG |
| EMDB ID | EMD-21875 | EMD-21882 | EMD-21892 |
| | Data collection/processing | | |
| Microscope | Talos Arctica | | |
| Camera | K3 | | |
| Magnification | 45,000X | | |
| Voltage (kV) | 200 | | |
| Total electron dose (e$^-$/Å$^2$) | 53 | | |
| Defocus range (μm) | −1.2 to −2.5 | | |
| Pixel size (Å) | 0.435 | | |
| Micrographs collected | 4525 | | |
| Final particles | 344069 | 139817 | 130240 |
| Symmetry | C1 | C1 | C1 |
| Resolution Å (FSC 0.143) | 2.8 | 3.1 | 3.2 |
| | Model composition | | |
| Non-hydrogen atoms | 21,310 | 26,835 | 25,932 |
| Protein residues | 2729 | 3450 | 3334 |
| Ligands | 0 | 6 | 5 |
| | Refinement | | |
| Map-model CC | 0.82 | 0.77 | 0.74 |
| RMSD bond lengths (Å) | 0.015 | 0.003 | 0.011 |
| RMSD bond angles (degrees) | 1.16 | 0.73 | 1.13 |
| | Validation | | |
| MolProbity score | 0.89 | 0.94 | 0.96 |
| Clash score | 1.5 | 1.9 | 1.7 |
| C-beta deviations | 0 | 0 | 0 |
| Rotamer outliers (%) | 0 | 0.07 | 0 |
| Ramachandran favored (%) | 98.9 | 99 | 99.3 |
| Ramachandran disallowed (%) | 0 | 0 | 0 |

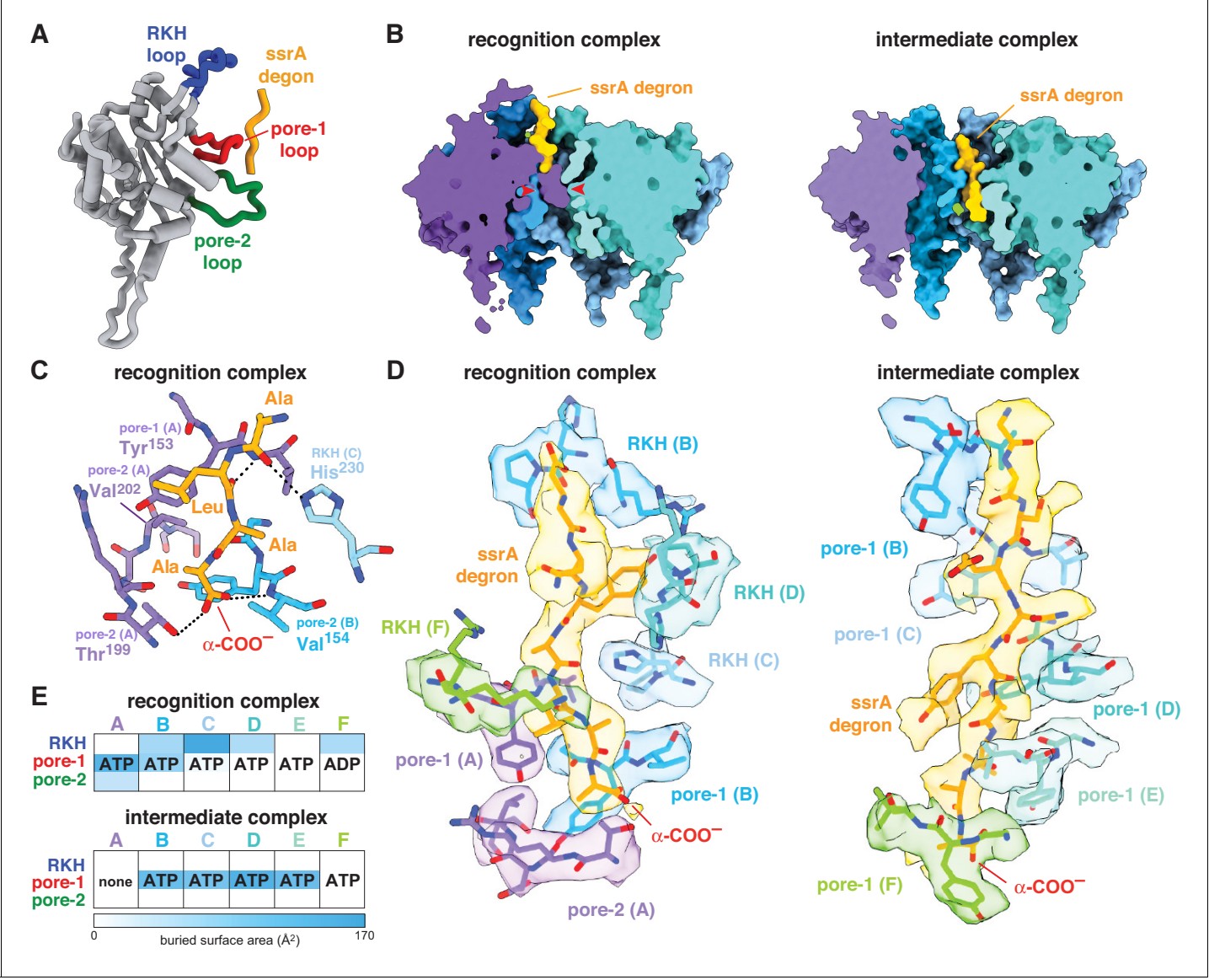

**Figure 2.** ClpX-degron interactions. (A) Positions of the RKH loop (blue), pore-1 loop (red), and pore-2 loop (green) in subunit A of the ClpX hexamer in the recognition complex relative to the positions of the ssrA degron (orange/gold). (B) Cutaway views of the recognition complex (left) and intermediate complex (right). In the recognition complex, the pore-2 loop of ClpX subunit A (denoted by red arrows) blocks the axial pore. The pore is open in the intermediate complex. (C) Key ClpX residues (blue or purple) and the C-terminal segment of the ssrA degron (gold) in the recognition complex are shown in stick representation. Dashed lines indicate hydrogen bonds. (D) ClpX-degron contacts in the recognition and intermediate complexes. ClpX or degron residues are shown in stick representation with semi-transparent density. (E) Subunit nucleotide state and degron/pore-loop interactions in the recognition and intermediate complexes.

The online version of this article includes the following figure supplement(s) for figure 2:

**Figure supplement 1.** Detailed contacts between ClpX and the ssrA degron.
**Figure supplement 2.** Nucleotide density in subunits of the recognition complex (top) and the intermediate complex (bottom).

supporting a key role for a hydrogen bond between the side-chain hydroxyl of Thr[199] in ClpX and the α-carboxylate of the ssrA degron (*Figure 2C*, *Figure 2—figure supplement 1A and B*). Third, compared to wild-type ClpXP, the R228A variant displays reduced specificity for an ssrA-tagged substrate and increased specificity for an N-degron substrate (*Farrell et al., 2007*). Finally, human ClpXP has leucines at positions corresponding to Thr[199] and His[230] in *E. coli* ClpX and does not degrade ssrA-tagged substrates, but a human hybrid containing transplanted pore-2 and RKH loops from the *E. coli* enzyme acquires this activity (*Martin et al., 2008b*).

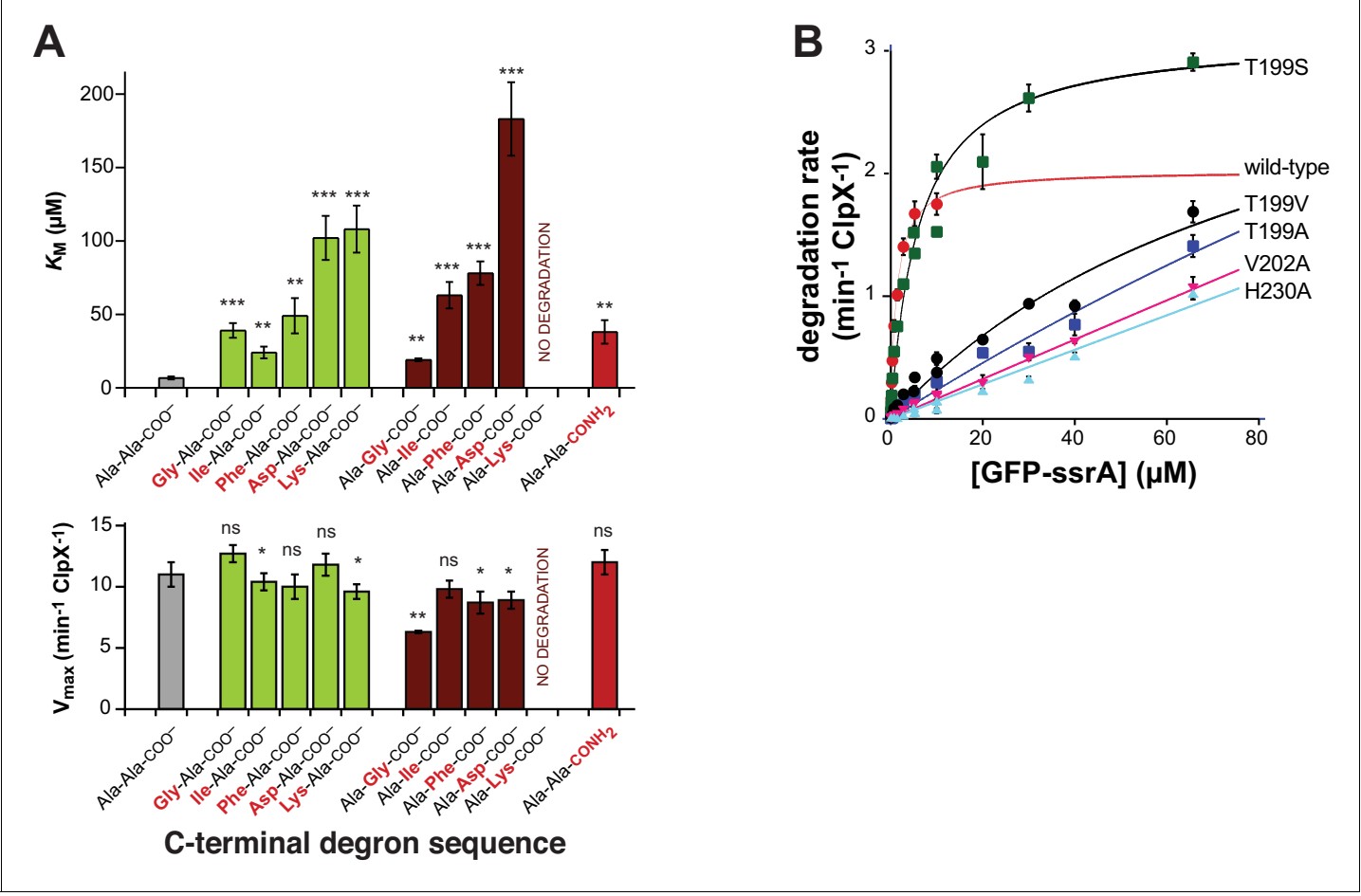

**Figure 3.** Mutations affecting recognition-complex contacts increase $K_M$ for ClpXP degradation. (A) Steady-state $K_M$ (top) and $V_{max}$ (bottom) parameters (means ± SD; n = 3) for wild-type ClpXP degradation of synthetic peptide substrates containing an aminobenzoic-acid (ABZ) fluorophore and nitrotyrosine ($Y^{NO2}$) quencher (wild-type sequence ABZ-FAPHMALVPY$^{NO2}$GYGGKKLAANDENYALAA; gray bar) with substitutions for the penultimate residue (green bars), C-terminal residue (dark red bars), or α-carboxylate (red bar). The statistical significance of $K_M$ or $V_{max}$ values relative to the wild-type Ala-Ala-COO$^-$ values was calculated using Student's two-tailed t-test (ns, not significant; *, p<0.05; **, p<0.01; ***, p<0.001). (B) Degradation of GFP-ssrA by ClpX$^{\Delta N}$/ClpP and variants. Data are means ± SD (n = 3 to 6), and lines are fits to the Michaelis-Menten equation. Fitted $K_M$ values (µM) were 1.3 ± 0.2 (wild type), 6.3 ± 1.7 (T199S), and 89 ± 34 µM (T199V). Fitted $V_{max}$ values (min$^{-1}$ ClpX$^{-1}$) were 2.0 ± 0.1 (wild type), 3.1 ± 0.2 (T199S), and 3.7 ± 2.0 (T199V). For the T199A, V202A, and H230A variants, unconstrained fits had huge error limits, and fits with $V_{max}$ constrained to less than or equal to 4 min$^{-1}$ ClpX$^{-1}$ gave $K_M$ values from 140 to 240 µM.

### The intermediate complex resembles translocation complexes

In the intermediate complex, five ClpX pore-1 loops and four pore-2 loops packed against degron side chains with a periodicity of two residues (*Figure 2B and C*, *Figure 2—figure supplement 1C and D*, *Video 2*). This arrangement of pore loops interacting with polypeptide in the channel has been observed previously in ClpXP complexes with other substrates and in different AAA+ proteases and protein-remodeling machines (*Fei et al., 2020*; *Ripstein et al., 2020*; *Puchades et al., 2020*). These enzyme structures are thought to reflect snapshots during non-specific translocation. Indeed, ClpXP translocates a variety of sequences, including polymeric tracts of glycine, proline, lysine, arginine, glutamate, and glutamine (*Barkow et al., 2009*), and the specific contacts observed in the recognition complex were absent in the intermediate complex. Non-specific translocation allows ClpXP to degrade any protein after degron recognition and unfolding of attached native structure (*Sauer and Baker, 2011*). By contrast, our results suggest that the pore-2 loop of ClpX subunit A occludes the axial channel (*Figure 2B*) prior to substrate binding, thereby preventing non-specific binding and degron-independent degradation.

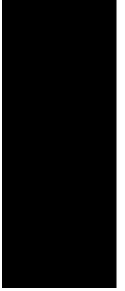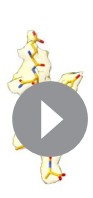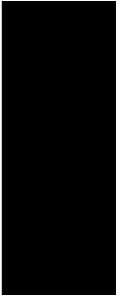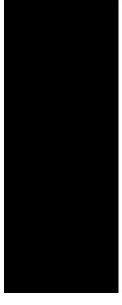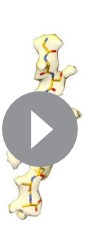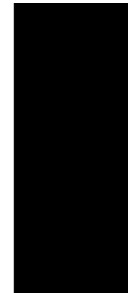

**Video 1.** Interaction of the ssrA degron with ClpX in the recognition complex. The ssrA tag is contacted by pore loops located at the top of the axial channel of ClpX. EM density is shown as a transparent surface. Hydrogen bonds between pore loops and the ssrA degron are shown as dashed lines.
https://elifesciences.org/articles/61496#video1

**Video 2.** Interaction of the ssrA degron with ClpX in the intermediate complex. The pore-1 and pore-2 loops from different subunits of ClpX interact with every two residues of the ssrA degron, as observed in other structures of ClpXP and related AAA+ proteins. EM density is shown as a transparent surface. Hydrogen bonds between pore loops and the ssrA degron are shown as dashed lines.
https://elifesciences.org/articles/61496#video2

*Table 2A* lists Cα RMSDs between ClpX hexamers in the intermediate or recognition complexes and prior high-resolution structures of ClpXP 'translocation' complexes (*Fei et al., 2020*; *Ripstein et al., 2020*). *Table 2B* shows that each of these structures contains ATP/ATPγS in subunits B, C, D, and E, with differences in nucleotide

**Table 2.** Comparisons between high-resolution ClpXP structures.
(A). RMSDs between Cα positions. (B). Nucleotides bound in different subunits of high-resolution ClpXP structures. Gray shading indicates subunits that always contain ATP or ATPγS.

**Table 2A**

| PDB ID | Name | Reference | RMSD (Å) Recognition complex | RMSD (Å) Intermediate complex |
|---|---|---|---|---|
| 6WSG | Intermediate complex | This paper | 2.8 | 0.0 |
| 6PP8 | Class 1 | *Fei et al., 2020* | 2.0 | 1.9 |
| 6PP7 | Class 2 | *Fei et al., 2020* | 2.8 | 1.3 |
| 6PP6 | Class 3 | *Fei et al., 2020* | 1.4 | 2.7 |
| 6PP5 | Class 4 | *Fei et al., 2020* | 1.4 | 2.9 |
| 6VFS | Conformation A | *Ripstein et al., 2020* | 3.0 | 2.6 |
| 6VFX | Conformation B | *Ripstein et al., 2020* | 1.6 | 3.0 |

**Table 2B**

| PDB ID | Name | ClpX subunit A | B | C | D | E | F | Reference |
|---|---|---|---|---|---|---|---|---|
| 6WRF | Recognition complex | ATP | ATP | ATP | ATP | ATP | ADP | This paper |
| 6WSG | Intermediate complex | - | ATP | ATP | ATP | ATP | ATP | This paper |
| 6PP8 | Class 1 | ATP | ATP | ATP | ATP | ATP | ADP | *Fei et al., 2020* |
| 6PP7 | Class 2 | ADP | ATP | ATP | ATP | ATP | ATP | *Fei et al., 2020* |
| 6PP6 | Class 3 | ATP | ATP | ATP | ATP | ATP | ADP | *Fei et al., 2020* |
| 6PP5 | Class 4 | ATP | ATP | ATP | ATP | ATP | ADP | *Fei et al., 2020* |
| 6VFS | Conformation A | ADP | ATP | ATP | ATP | ATP | ADP | *Ripstein et al., 2020* |
| 6VFX | Conformation B | ATP | ATP | ATP | ATP | ATP | ADP | *Ripstein et al., 2020* |

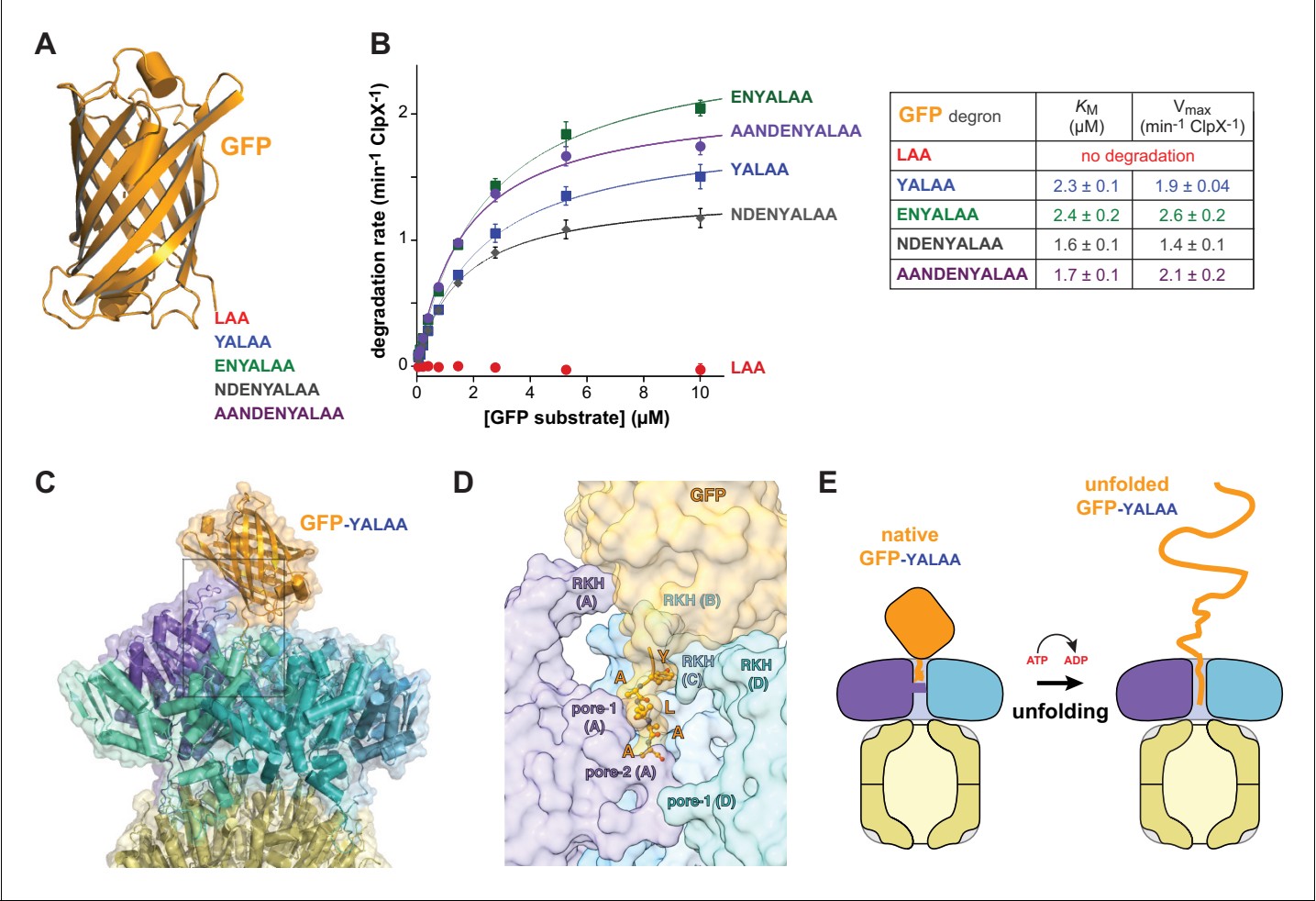

**Figure 4.** Degradation dependence on degron length. (**A**) Cartoon of GFP (pdb 1EMA; *Ormö et al., 1996*) with degron tails of 3, 5, 7, 9, and 11 residues. (**B**) Plots of steady-state ClpX$^{\Delta N}$/ClpP degradation rates (means ± SD; n = 3) as a function of substrate concentration. Lines are fits to the Michaelis-Menten equation. Fitted $K_M$ and $V_{max}$ values represent averages of three independent determinations ± SD. (**C**) Model of GFP-YALAA bound to ClpXP (cartoon and semi-transparent surface representation) created by aligning the YALAA of GFP to the same sequence in the recognition complex and then minimizing clashes both manually and computationally. (**D**) Close-up view of the ClpX-GFP-YALAA model (subunits E and F removed), corresponding to the dashed rectangle in panel C, with the YALAA shown in ball-and-stick representation. (**E**) Cartoon in which one power stroke unfolds GFP by translocating the YALAA degron six residues deeper into the channel of ClpX.

occupancy of subunits A and F. Among these structures, the intermediate complex is unique in containing a nucleotide-free subunit A.

## Dependence of substrate unfolding on degron length

Can a stable protein substrate, like GFP, be unfolded directly from the recognition complex or is substrate engagement by additional pore-1 and pore-2 loops deeper in the ClpX channel, as a consequence of one or more translocation steps, required to allow mechanical substrate denaturation? If direct unfolding from the recognition complex is possible, then a degron of ~5 residues, which is the number of ssrA-tag residues interacting with ClpX in the recognition complex, should be sufficient for degradation. By this model, longer degrons should also support degradation, but shorter degrons should not because they cannot make contacts needed for recognition. To test this model, we constructed substrates with degrons of 3, 5, 7, 9, or 11 residues following the last structured residue of GFP (*Figure 4A*) and assayed degradation.

Strikingly, GFP-LAA was not degraded, whereas GFP-YALAA and substrates with longer tags were robustly degraded (*Figure 4B*). Modeling revealed that the native barrel of GFP-YALAA docked snugly with the top of the AAA+ ClpX ring, with the YALAA in the same position as in the recognition

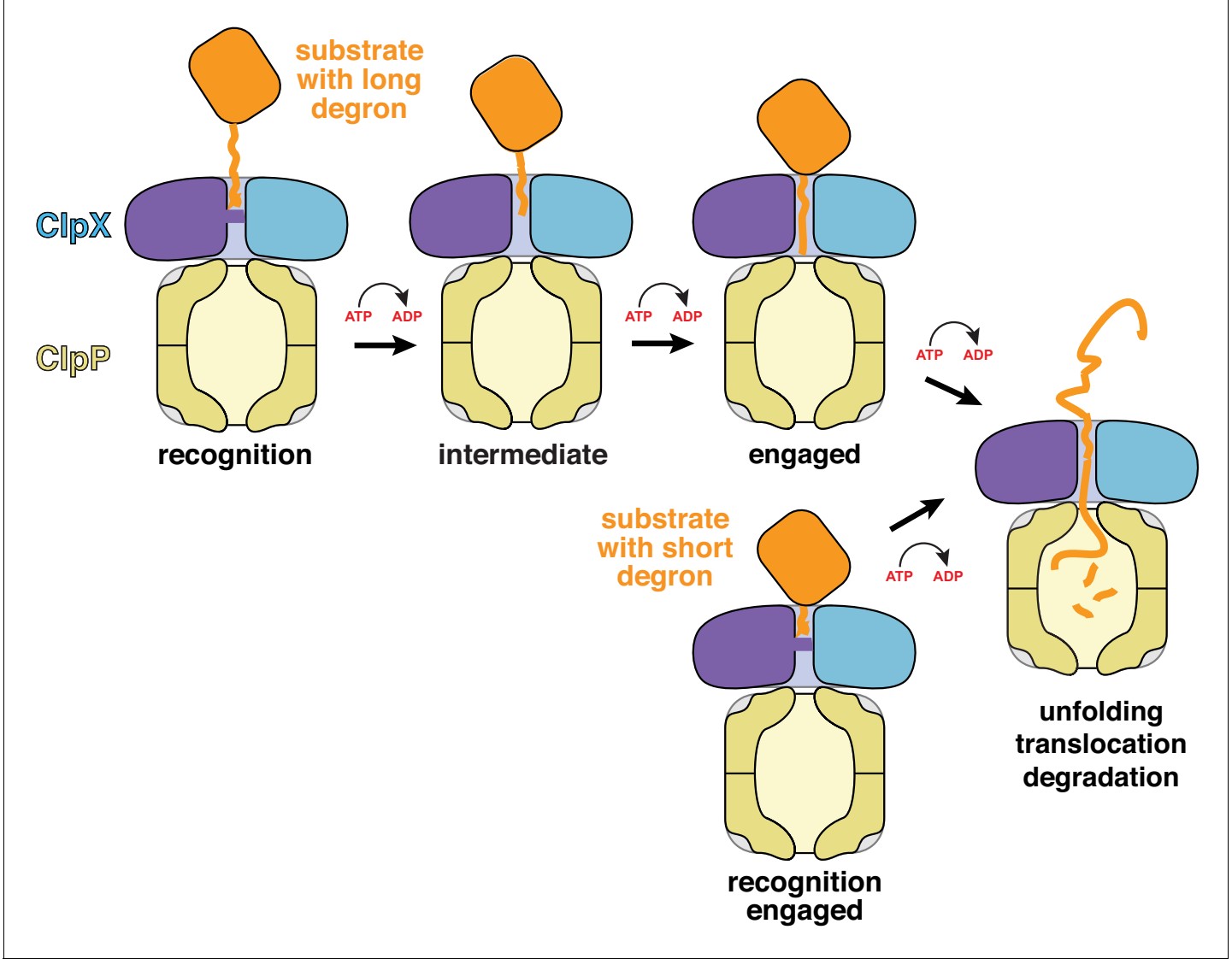

**Figure 5.** Models for substrate recognition, engagement, and unfolding by ClpXP. Top. A substrate with a relatively long degron (~20 residues) is recognized and subsequent ATP-dependent power strokes then move the degron deeper into the ClpX channel in the intermediate complex, and then the engaged complex, from which unfolding occurs. Bottom. A substrate with a short degron (~5 residues) forms a recognition complex that is engaged and can therefore carry out direct ATP-dependent unfolding.

complex (*Figure 4C and D*; *Video 3*). By contrast, the tag of GFP-ʟᴀᴀ was too short to allow formation of recognition-complex contacts without severe steric clashes with ClpX. It might be argued that the C-terminal β-strand of GFP-ʏᴀʟᴀᴀ, which is ~20 residues in length, unfolds to allow additional C-terminal residues of the substrate to bind deeper in the channel of ClpX. However, global GFP unfolding occurs with a half-life of years (*Kim et al., 2000*), and the C-terminal β-strand remains stably associated even when it is non-covalently attached to the remaining native structure (*Nager et al., 2011*). The inability of ClpXP to degrade GFP-ʟᴀᴀ also argues against a model for degradation in which the C-terminal β-strand of GFP spontaneously denatures. We conclude that a power stroke initiated directly from the recognition complex can unfold GFP-ʏᴀʟᴀᴀ (*Figure 4E*).

Subunits ABCDE of ClpX contained ATP/ATPγS and subunit F contained ADP in the recognition complex, whereas subunit A was nucleotide free and the remaining subunits were ATP/ATPγS bound in the intermediate complex (*Figure 2C*, *Figure 2—figure supplement 2*). Thus, it is plausible that ATP hydrolysis in subunit A of the recognition complex drives an unfolding power stroke and conversion to the intermediate complex. *Video 4* shows a morph illustrating the transition between these

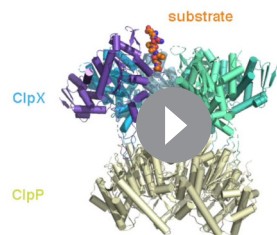

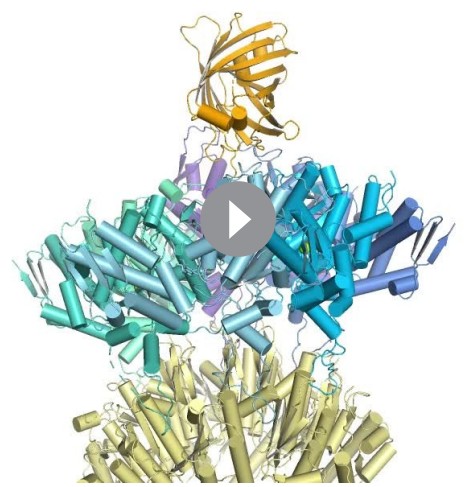

**Video 3.** Model of the interaction between GFP-ʏᴀʟᴀᴀ and the ClpX portion of the recognition complex. GFP with a five-residue ʏᴀʟᴀᴀ degron docks snuggly and without major clashes with the ClpX ring.
https://elifesciences.org/articles/61496#video3

**Video 4.** Side and top views of a morph between the recognition and intermediate complexes with one ClpX subunit removed for clarity. This morph was generated by aligning subunit A in the recognition complex spiral with subunit F in the intermediate complex spiral after superimposing the ClpP portions of the two structures. In the morph, the purple subunit and ssrA degron move 25 Å or six residues deeper into the axial channel toward ClpP.
https://elifesciences.org/articles/61496#video4

states. The six-residue offset in the position of the ssrA tag between the two complexes (*Figure 1B*) is consistent with optical-trap experiments that show a fundamental ClpXP translocation step of 5–8 amino acids (*Aubin-Tam et al., 2011*; *Maillard et al., 2011*). Hence, for sufficiently short tags, a single power stroke from the recognition complex could move the tag six residues deeper into the channel, unfolding native structure N-terminal to the tag in the process (*Figures 4E* and *5*, bottom pathway). For longer tags, including the 20-residue degron used for our cryo-EM studies, continued translocation after the first power stroke would eventually bring the native protein in contact with the top of the ClpX channel, where additional power strokes could apply an unfolding force (*Figure 5*, top pathway).

## Discussion

Our cryo-EM structures of the recognition and intermediate complexes represent the first instances in which ClpX interactions with specific side chains of a protein substrate have been visualized. The general properties of the intermediate complex are similar to previously determined structures of ClpXP translocation complexes in the sense that the axial channel is open and five pore-1 loops and multiple pore-2 loops contact every two residues of the substrate polypeptide in the channel (*Fei et al., 2020*; *Ripstein et al., 2020*). This structural feature has been widely observed in AAA+ proteases and protein-remodeling machines, suggesting that these diverse molecular machines employ a common mechanism of substrate translocation (for review, see *Puchades et al., 2020*).

Strikingly, the recognition complex of ClpX bound to the ssrA degron is unique compared to previously determined structures. For example, all previous structures have open channels, whereas the axial channel of the AAA+ ring of ClpX in the recognition complex is closed by the pore-2 loop of subunit A, which makes specific contacts with the C-terminal residue of the ssrA tag. Moreover, only the pore-1 loops of the top two subunits in the ClpX spiral contact the substrate degron in the recognition complex, as opposed to contacts between five pore-1 loops and substrates in other known structures. The contacts we see between ClpX and the ssrA degron in the recognition complex explain multiple biochemical results, including ones that show that the C-terminal Ala-Ala dipeptide of the ssrA tag is most important for ClpXP degradation (*Flynn et al., 2001*; *Figure 3A*) and others that demonstrate that the side chains of six ClpX residues in the pore-1 loop (Tyr$^{153}$, Val$^{154}$), pore-2 loop (Thr$^{199}$, Val$^{202}$), and RKH loop (Arg$^{228}$, His$^{230}$) play critical roles both in substrate binding and in

substrate specificity (*Siddiqui et al., 2004*; *Farrell et al., 2007*; *Martin et al., 2008b*; *Iosefson et al., 2015*; *Figure 3B*).

In addition to providing a specific binding site for the ssrA degron, the closed axial channel of the recognition complex would preclude the non-specific binding of substrate polypeptides that is observed in the open channels of the intermediate complex and related structures. Non-specific binding in the intermediate complex explains how ClpXP efficiently translocates and degrades diverse polypeptide sequences (*Barkow et al., 2009*), including unnatural amino acids, but could also result in inappropriate degradation of proteins that initially bound in an open channel. Our structures suggest a simple solution to this problem. Only following specific recognition of the ssrA degron in the closed-pore conformation does the pore-2 loop of subunit A move to permit passage and interaction of deeper channel residues with any polypeptide sequence.

Structures, by themselves, can suggest but do not establish order in a kinetic pathway. However, a recent study of the kinetics of ClpXP association with a substrate bearing a 20-residue ssrA degron similar to the one studied here provides evidence for three sequentially occupied substrate-bound conformations (*Saunders et al., 2020*). The first two are likely to correspond to our recognition and intermediate complexes. Conversion of the first to the second complex depends on the rate of ATP hydrolysis, as does formation of the third kinetically defined state, which is probably similar to the fully engaged complex depicted in *Figure 5*. Prior ClpXP complexes showing the native portion of a protein substrate contacting the top of the central channel with an attached peptide filling the channel (*Fei et al., 2020*) provide structural evidence for this fully engaged state. Our results indicate that ClpX can unfold short-degron substrates directly from the closed-channel recognition complex, in which just two pore-1 loops contact the degron, and long-degron substrates from a subsequent open-channel engaged complex in which five pore-1 loops engage the substrate (*Figure 5*). Although unfolding in these two cases must be somewhat different in terms of the detailed mechanism, substrate residues in the upper part of the channel are gripped most tightly during ClpXP unfolding of substrates with long degrons (*Bell et al., 2019*), supporting a model in which ClpX-substrate contacts near the top of the channel play key roles in both unfolding mechanisms.

The multistep mechanism proposed here for protein degradation by ClpXP may be used by related AAA+ proteases that recognize unstructured degrons, as it nicely resolves how unfolding motors make initiation highly specific (and thus limited to appropriate protein targets) but then allow nonspecific unfolding and translocation for subsequent degradation and/or remodeling. Whether other AAA+ enzymes also use closed axial channels for degron recognition remains to be determined. More generally, AAA+ proteases and protein-remodeling machines must deal with problems similar to those encountered by multisubunit enzymes in transcription, translation, DNA replication, and protein secretion, which also require transitions from specific recognition conformations to complexes that utilize the chemical energy of ATP or GTP hydrolysis to processively and non-specifically move along their polymeric substrates.

## Materials and methods

**Key resources table**

| Reagent type (species) or resource | Designation | Source or reference | Identifiers | Additional information |
|---|---|---|---|---|
| Strain, strain background (*Escherichia coli*) | ER2566 | NEB | 1312 | Chemically competent cells |
| Gene (*Escherichia coli*) | clpX | *E. coli* (strain K12) EXPASY | UniProtKB- P0A6H1 | |
| Gene (*Escherichia coli*) | clpP | *E. coli* (strain K12) EXPASY | UniProtKB- P0A6G7 | |
| Recombinant DNA reagent | pT7 ClpX (plasmid) | *Kim et al., 2000* | | N-terminally His$_6$-tagged ClpX for overexpression |

*Continued on next page*

*Continued*

| Reagent type (species) or resource | Designation | Source or reference | Identifiers | Additional information |
|---|---|---|---|---|
| Recombinant DNA reagent | pT7 ClpX$^{\Delta N}$(plasmid) | *Martin et al., 2005* | | N-terminally His$_6$-tagged ClpX$^{\Delta N}$ (residues 62–424) for overexpression |
| Recombinant DNA reagent | pT7-ClpX$^{\Delta N}$-T199A (plasmid) | This paper, Material and methods | | ClpX$^{\Delta N}$ (residues 62–424) T199A mutant, can be obtained from the Sauer lab |
| Recombinant DNA reagent | pT7-ClpX$^{\Delta N}$-T199S (plasmid) | This paper, Material and methods | | ClpX$^{\Delta N}$ (residues 62–424) T199S mutant, can be obtained from the Sauer lab |
| Recombinant DNA reagent | pT7-ClpX$^{\Delta N}$-T199V (plasmid) | This paper, Material and methods | | ClpX$^{\Delta N}$ (residues 62–424) T199V mutant, can be obtained from the Sauer lab |
| Recombinant DNA reagent | pT7-ClpX$^{\Delta N}$-V202A (plasmid) | This paper, Material and methods | | ClpX$^{\Delta N}$ (residues 62–424) V202A mutant, can be obtained from the Sauer lab |
| Recombinant DNA reagent | pT7-ClpX$^{\Delta N}$-H230A (plasmid) | This paper, Material and methods | | ClpX$^{\Delta N}$ (residues 62–424) H230A mutant, can be obtained from the Sauer lab |
| Recombinant DNA reagent | PACYC-ClpX$^{\Delta N}_6$-TEV-cHis$_6$ (plasmid) | *Martin et al., 2005* | | ClpX expression, can be obtained from the Sauer lab |
| Recombinant DNA reagent | pT7 ClpP (plasmid) | *Kim et al., 2000* | | C-terminally His$_6$-tagged ClpP for overexpression |
| Recombinant DNA reagent | pT7-GFP-ssrA (plasmid) | *Kim et al., 2000* | | Expresses fluorescent substrate for degradation assays, can be obtained from the Sauer lab |
| Recombinant DNA reagent | pT7 GFP LAA (plasmid) | This paper | | N-terminally His$_6$-tagged GFP (1-229) substrates with a LAA C-terminal tail, for overexpression. |
| Recombinant DNA reagent | pT7 GFP YALAA (plasmid) | This paper | | N-terminally His$_6$-tagged GFP (1-229) substrates with a YALAA C-terminal tail, for overexpression. |
| Recombinant DNA reagent | pT7 GFP ENYALAA (plasmid) | This paper | | N-terminally His$_6$-tagged GFP (1-229) substrates with a ENYALAA C-terminal tail, for overexpression. |

*Continued on next page*

*Continued*

| Reagent type (species) or resource | Designation | Source or reference | Identifiers | Additional information |
|---|---|---|---|---|
| Recombinant DNA reagent | pT7 GFP NDENYALAA (plasmid) | This paper | | N-terminally His$_6$-tagged GFP (1-229) substrates with a NDENYALAA C-terminal tail, for overexpression. |
| Recombinant DNA reagent | pT7 GFP AANDENYALAA (plasmid) | This paper | | N-terminally His$_6$-tagged GFP (1-229) substrates with a AANDENYALAA C-terminal tail, for overexpression. |
| Peptide, recombinant protein | ABZ-FAPHMAL VPY$^{NO2}$GYGGKK LAANDE NYALAA-COO$^-$ | This paper | | Fluorescent peptide, for degradation assays |
| Peptide, recombinant protein | ABZ-FAPHMAL VPY$^{NO2}$GYGGKKLAAN DENYALGA-COO$^-$ | This paper | | Fluorescent peptide, for degradation assays |
| Peptide, recombinant protein | ABZ-FAPHMAL VPY$^{NO2}$GYGGKKLAAN DENYALIA-COO$^-$ | This paper | | Fluorescent peptide, for degradation assays |
| Peptide, recombinant protein | ABZ-FAPHMAL VPY$^{NO2}$GYGGKKLAAN DENYALFA-COO$^-$ | This paper | | Fluorescent peptide, for degradation assays |
| Peptide, recombinant protein | ABZ-FAPHMAL VPY$^{NO2}$GYGGKKLAAN DENYALDA-COO$^-$ | This paper | | Fluorescent peptide, for degradation assays |
| Peptide, recombinant protein | ABZ-FAPHMAL VPY$^{NO2}$GYGG KKLAAN DENYALKA-COO$^-$ | This paper | | Fluorescent peptide, for degradation assays |
| Peptide, recombinant protein | ABZ-FAPHMAL VPY$^{NO2}$GYGGK KLAAN DENYALAG-COO$^-$ | This paper | | Fluorescent peptide, for degradation assays |
| Peptide, recombinant protein | ABZ-FAPHMAL VPY$^{NO2}$GYGGK KLAAN DENYALAI-COO$^-$ | This paper | | Fluorescent peptide, for degradation assays |
| Peptide, recombinant protein | ABZ-FAPHMAL VPY$^{NO2}$GYGGK KLAAN DENYALAF-COO$^-$ | This paper | | Fluorescent peptide, for degradation assays |
| Peptide, recombinant protein | ABZ-FAPHMAL VPY$^{NO2}$GYGGK KLAAN DENYALAD-COO$^-$ | This paper | | Fluorescent peptide, for degradation assays |
| Peptide, recombinant protein | ABZ-FAPHMAL VPY$^{NO2}$GYGGK KLAAN DENYALAK-COO$^-$ | This paper | | Fluorescent peptide, for degradation assays |
| Peptide, recombinant protein | ABZ-FAPHMAL VPY$^{NO2}$GYGGK KLAAN DENYALAA-CONH$_2$ | This paper | | Fluorescent peptide, for degradation assays |
| Software, algorithm | Relion | *Zivanov et al., 2018* | RRID:SCR_016274 | EM reconstruction software |

*Continued on next page*

*Continued*

| Reagent type (species) or resource | Designation | Source or reference | Identifiers | Additional information |
|---|---|---|---|---|
| Software, algorithm | UCSF Chimera | *Pettersen et al., 2004* | RRID:SCR_004097 | Molecular visualization software |
| Software, algorithm | UCSF ChimeraX | *Goddard et al., 2018* | RRID:SCR_015872 | Molecular visualization software |
| Software, algorithm | Phenix | *Adams et al., 2010* | RRID:SCR_014224 | Structure refinement software |
| Software, algorithm | MolProbity | *Williams et al., 2018* | RRID:SCR_014226 | Protein modele valuation software |
| Software, algorithm | PyMOL | Schrödinger, LLC. | RRID:SCR_000305 | Molecular visualization software |
| Software, algorithm | Coot | *Emsley and Cowtan, 2004* | RRID:SCR_014222 | Protein model building software |
| Software, algorithm | Ctffind | *Rohou and Grigorieff, 2015* | RRID:SCR_016732 | EM image analysis software |
| Software, algorithm | PISA | 'Protein interfaces, surfaces and assemblies' service PISA at the European Bioinformatics Institute (http://www.ebi.ac.uk/pdbe/prot_int/pistart.html) | RRID:SCR_015749 | Protein modelanalysis software |

## Proteins and peptides

His$_6$-tagged variants of *E. coli* ClpX, ClpX$^{\Delta N}$ (residues 62–424), single-chain ClpX$^{\Delta N}$ (units connected by six-residue linkers), and ClpP were purified as described (*Kim et al., 2000*; *Bell et al., 2019*). Peptide substrates were synthesized as described (*Barkow et al., 2009*), and mutant variants of ClpX$^{\Delta N}$ or GFP were constructed by PCR as described (*Stinson et al., 2013*).

## Cryo-EM

We combined single-chain ClpX$^{\Delta N}$ (4 μM), ClpP (1.8 μM), and ATPγS (2.5 mM) in buffer containing 20 mM HEPES (pH 7.5), 100 mM KCl, and 25 mM MgCl$_2$. After 5 min at 25°C, we added an equal volume of GFP-$_{G_3YG_9SENYALAA}$ (20 μM) and ATP (2.5 mM) in the same buffer, and applied 3 μL to glow discharged EM grids (Quantifoil R2/2 400 mesh). The GFP portion of this substrate consisted of residues 1–229, which includes only the C-terminal amino acids observed in a crystal structure (*Ormö et al., 1996*). An ATPγS/ATP mixture was used to allow rapid substrate binding, while slowing translocation and unfolding (*Burton et al., 2003*; *Martin et al., 2008a*; *Saunders et al., 2020*). Samples were blotted and vitrified within 15 s of final mixing. 4525 micrographs were collected using a Talos Arctica with a Gatan K3 direct electron detector (super-resolution pixel 0.435 Å; uncalibrated magnification X45,000). EM images contained complexes in which ClpP$_{14}$ was bound to one (~60%) or two (~40%) ClpX$^{\Delta N}$ hexamers. After three rounds of 2D classification, 344,069 complexes with one ClpX$^{\Delta N}$ hexamer were selected for 3D reconstruction. Using a 40 Å low-pass filtered ClpP map as the search model (EMDB: EMD-20434, *Fei et al., 2020*) but no mask, 3D auto-refinement without symmetry (C1) yielded a ClpXP map at 3.5 Å resolution. CTF-refinement and particle polishing improved the overall resolution of this map to 2.8 Å. After focused classification on ClpX without alignment, multiple runs with different class numbers converged to two major classes, which were later named the recognition complex (3.1 Å resolution) and intermediate complex (3.2 Å resolution).

Data analysis and reconstruction was performed within the Relion 3.0.8 pipeline (*Zivanov et al., 2018*). We docked structures of ClpP (pdb 6PPE) or ClpXP-substrate complexes (pdb 6PP5 and 6PP7) into EM maps using Chimera (*Pettersen et al., 2004*), rigid-body refined ClpX domains using Coot (*Emsley and Cowtan, 2004*), and performed real-space refinement using PHENIX (*Adams et al., 2010*). The ssrA degron was first modeled and refined as polyalanine, and specific side chains were modeled and refined subsequently.

### Degradation assays

Peptide degradation by wild-type ClpXP (0.3 μM full-length ClpX$_6$; 0.8 μM ClpP$_{14}$) was assayed by changes in fluorescence (excitation/emission: 320/420 nm) at 30°C in 25 mM HEPES [pH 7.6], 200 mM KCl, 5 mM MgCl$_2$, 0.032% NP-40, 10% glycerol with 4 mM ATP, 16 mM creatine phosphate, and 0.32 mg/ml creatine phosphokinase (*Barkow et al., 2009*). Degradation of ssrA-tagged GFP substrates by wild-type or variant ClpX$^{\Delta N}$ hexamers (0.1–0.3 μM) and ClpP$_{14}$(0.3–0.9 μM) was assayed by changes in fluorescence (excitation/emission: 467/511 nm for low-substrate concentrations or 420/540 nm for high substrate concentrations) under similar conditions but at 37°C, without NP-40, and with 5 mM ATP (*Kim et al., 2000*).

To model GFP-YALAA binding to ClpXP, we extended the C-terminal β-strand of GFP (pdb 1EMA; residues 2–229) with the sequence YALAA and aligned this sequence with the YALAA of recognition-complex degron using the PyMOL Molecular Graphics System, v. 2.0 (Schrödinger, LLC). Analysis in MolProbity (*Williams et al., 2018*) revealed multiple clashes between GFP and the RKH loops of ClpX, which were minimized manually in Coot (*Emsley and Cowtan, 2004*) by rotation of the residue 2–229 segment of GFP relative to the rest of the complex. To minimize clashes further, we created a map of the modeled complex and optimized geometry using the Calculate F(model) and Real-space refinement utilities, respectively in PHENIX (*Adams et al., 2010*). The final model had a MolProbity score of 1.0 (100th percentile) and one minor clash (0.44 Å) between GFP and ClpX.

## Acknowledgements

Supported by NIH grant GM-101988. We thank C Xu, K Song, and K Lee at the Electron Microscopy Facility at the U Mass. Medical School for advice and data collection and S Jenni and D Southworth for thoughtful comments on the manuscript.

## Additional information

### Funding

| Funder | Grant reference number | Author |
| --- | --- | --- |
| National Institute of General Medical Sciences | GM-101988 | Robert T Sauer |

The funders had no role in study design, data collection and interpretation, or the decision to submit the work for publication.

### Author contributions

Xue Fei, Conceptualization, Data curation, Validation, Investigation, Visualization, Methodology, Writing - original draft, Writing - review and editing; Tristan A Bell, Conceptualization, Data curation, Formal analysis, Investigation, Writing - original draft, Writing - review and editing; Sarah R Barkow, Investigation, Writing - review and editing; Tania A Baker, Supervision, Funding acquisition, Writing - review and editing; Robert T Sauer, Supervision, Funding acquisition, Investigation, Writing - original draft, Writing - review and editing

### Author ORCIDs

Tania A Baker http://orcid.org/0000-0002-0737-3411
Robert T Sauer https://orcid.org/0000-0002-1719-5399

**Decision letter and Author response**
Decision letter https://doi.org/10.7554/eLife.61496.sa1
Author response https://doi.org/10.7554/eLife.61496.sa2

## Additional files

### Supplementary files

• Transparent reporting form

### Data availability

Structures have been deposited in PDB under the accession codes 6WSG, 6WR2, and 6WRF.

The following datasets were generated:

| Author(s) | Year | Dataset title | Dataset URL | Database and Identifier |
| --- | --- | --- | --- | --- |
| Fei X, Sauer RT | 2020 | ClpXP-ssrA Intermediate complex | https://www.rcsb.org/structure/6WSG | RCSB Protein Data Bank, 6WSG |
| Fei X, Sauer RT | 2020 | ClpP | https://www.rcsb.org/structure/6WR2 | RCSB Protein Data Bank, 6WR2 |
| Fei X, Sauer RT | 2020 | ClpXP-ssrA Recognition complex | https://www.rcsb.org/structure/6WRF | RCSB Protein Data Bank, 6WRF |

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
