## [Decision Letter]

**Acceptance summary:**

This is the first high resolution structure of a substrate bound to the bacterial ClpXP degradation machinery. It uses cryoEM to reveal how the ssrA tag is recognised by ClpX's AAA+ ring.

**Decision letter after peer review:**

Thank you for submitting your article "Structural basis of ClpXP recognition and unfolding of ssrA-tagged substrates" for consideration by *eLife*. Your article has been reviewed by three peer reviewers, one of whom is a member of our Board of Reviewing Editors, and the evaluation has been overseen by Cynthia Wolberger as the Senior Editor. The reviewers have opted to remain anonymous.

The reviewers have discussed the reviews with one another and the Reviewing Editor has drafted this decision to help you prepare a revised submission.

Summary:

In *E. coli*, the AAA+ ClpXP protease degrades incomplete proteins resulting from stalled translation, which are labeled with a ssrA degron. Translocation of substrates by the ClpX AAA+ ring into the ClpP protease is non-sequence-specific, allowing the degradation of any degron-labeled incomplete protein. At the same time, specific recognition of the degron prevents the degradation of proteins that have not been labeled. How this initial sequence-specific recognition happens without allowing non-sequence-specific translocation, and how transitions to translocation, was not understood mechanistically. In this manuscript, Fei, Bell and colleagues use cryo-EM to solve the structures of two early stages in the process of engagement of substrates by ClpXP: (1) A "recognition" complex that shows how ClpXP binds to the ssrA degron in a sequence-specific manner, and in a conformation that prevents the non-specific engagement needed to lead to translocation and unfolding; and (2) an "intermediate" complex where the peptide has moved further into ClpXP, and where the interactions with the ssrA-substrate are reminiscent of previously reported translocation complexes that no longer engage substrates in a sequences-specific manner. The authors then use biochemistry to validate several of the sequence-specific interactions identified in the recognition complex. They also test the length requirements for the degron based on that structure.

Essential revisions:

1) More details on the cryo-EM data must be presented before the manuscript is ready for publication. Specifically:

a) The authors should include a supplementary figure with local resolution maps for the two structures. The images in Figure 1A suggest that ClpX is at a lower resolution than ClpP so it would be important for readers to know what this difference is more quantitatively. That figure should also include close-ups of different parts of the map spanning the resolution range, with the model fitted in. Similarly, if the maps in Figure 1A were locally filtered for display, that should be stated in the legend.

b) More generally, a supplementary figure on cryo-EM data quality and processing strategy would be nice. This would be the type of figure that shows the full processing pipeline (2D and 3D), showing the different structures that were obtained in 3D classification, and how many particles were chosen for the next step at each step. We are still at a point in the cryo-EM field where this information is both useful and interesting to other cryo-EM practitioners.

c) The authors must show close-ups of the cryo-EM map for all the nucleotide-binding sites listed in Figure 2C, with the model fitted in. Readers cannot be expected to accept the information in the table at face value.

d) The authors have submitted their cryo-EM maps. Please could they also submit the corresponding pdb files with the re-submission.

2) The authors should explain in more detail how the current work differs from and advances on the paper they published in *ELife* earlier in the year. The main text should contain details including the use of a defined substrate (vs. a non-specific peptide picked up during purification), the higher resolution of these current structures, a comparison of the previous structure with the intermediate form reported here etc. This would help a general reader to understand the advances in this work.

3) Some statistics for Figure 3A, indicating what differences are and are not significant, would be useful.

---

## [Author Response]

Essential revisions:1) More details on the cryo-EM data must be presented before the manuscript is ready for publication. Specifically:a) The authors should include a supplementary figure with local resolution maps for the two structures. The images in Figure 1A suggest that ClpX is at a lower resolution than ClpP so it would be important for readers to know what this difference is more quantitatively. That figure should also include close-ups of different parts of the map spanning the resolution range, with the model fitted in. Similarly, if the maps in Figure 1A were locally filtered for display, that should be stated in the legend.

We have added the requested local-resolution figure as part of Figure 1—figure supplement 2.

We have also added close-ups of different parts of the map as requested as part of Figure 1—figure supplement 3.

The maps were not locally filtered for display.

b) More generally, a supplementary figure on cryo-EM data quality and processing strategy would be nice. This would be the type of figure that shows the full processing pipeline (2D and 3D), showing the different structures that were obtained in 3D classification, and how many particles were chosen for the next step at each step. We are still at a point in the cryo-EM field where this information is both useful and interesting to other cryo-EM practitioners.

We have added the requested information as parts of Figure 1—figure supplements 1 and 2.

c) The authors must show close-ups of the cryo-EM map for all the nucleotide-binding sites listed in Figure 2C, with the model fitted in. Readers cannot be expected to accept the information in the table at face value.

As requested, we have added close-ups for density for all the nucleotide binding sites in the recognition and intermediate complexes as Figure 2—figure supplement 2.

d) The authors have submitted their cryo-EM maps. Please could they also submit the corresponding pdb files with the re-submission.

We apologize. The pdb files were uploaded to the *eLife* website, but obviously didn't make it to the reviewers.

2) The authors should explain in more detail how the current work differs from and advances on the paper they published in ELife earlier in the year. The main text should contain details including the use of a defined substrate (vs. a non-specific peptide picked up during purification), the higher resolution of these current structures, a comparison of the previous structure with the intermediate form reported here etc. This would help a general reader to understand the advances in this work.

Early in the Results section of the revised manuscript, we now write "The presence of clear side-chain density for the ssrA degron is the most important difference between our current structures and those previously reported by our group (Fei et al., 2020) and Ripstein et al. (2020)." We did not comment on higher resolution, as suggested, because the resolutions of the structures of Ripstein et al. were comparable to our current structures.

3) Some statistics for Figure 3A, indicating what differences are and are not significant, would be useful.

As requested, we have revised Figure 3A to contain significance probabilities calculated by Student's two-tailed t-test for comparisons of V_max_ or K_M_ values to the wild-type values.